# CDeepEx: Contrastive Deep Explanations

## Abstract

We propose a method which can *visually* explain the classification decision of deep neural networks (DNNs). There are many proposed methods in machine learning and computer vision seeking to clarify the decision of machine learning black boxes, specifically DNNs. All of these methods try to gain insight into why the network "chose class A" as an answer. Humans, when searching for explanations, ask two types of questions. The first question is, "Why did you choose this answer?" The second question asks, "Why did you *not* choose answer B over A?" The previously proposed methods are either not able to provide the latter directly or efficiently.

We introduce a method capable of answering the second question both directly and efficiently. In this work, we limit the inputs to be images. In general, the proposed method generates explanations in the input space of *any* model capable of efficient evaluation and gradient evaluation. We provide results, showing the superiority of this approach for gaining insight into the inner representation of machine learning models.

## 1 Introduction

Deep neural networks (DNN) have shown extraordinary performance on computer vision tasks such as image classification (Szegedy et al., 2015; Simonyan and Zisserman, 2014; Springenberg et al., 2015; He et al., 2016), image segmentation (Chen et al., 2018), and image denoising (Zhang et al., 2017). The first example of such a performance was on image classification, where it outperformed other computer vision methods which were carefully handcrafted for image classification (Krizhevsky et al., 2012). Following this success, DNNs continued to grow in popularity. Although the performances of DNNs on different tasks are getting close to human expertise (Rajpurkar et al., 2017) and in some cases surpass them (Springenberg et al., 2015), there is hesitation to use them when interpretability of the results is important. Accuracy is a well-defined criterion but does not provide useful understandings of the actual inner workings of the network. If the deployment of a network may result in inputs whose distribution differs from that of the training or testing data, interpretability or explanations of the network's decisions can be important for securing human trust in the network.

Explanations are important in settings such as medical treatments, system verification, and human training and teaching. Naturally, one way of getting an explanation is asking the *direct* question, "Why did the DNN choose this answer?" Humans often also seek *contrasting* explanations. For instance, they maybe more familiar with the contrasting answer, or they want to find the subtle differences in input which change the given answer to the contrasting one. This way of questioning can be phrased as, "Why did the DNN *not* choose B (over A)?" In this work, we present a framework to answer this type of question.

We learn a model over the input space which is capable of generating synthetic samples similar to the input. Then, we ask how we can alter this *synthetic* input to change the classification outcome. Our proposed framework is not based on heuristics, does not need to change the given network, is applicable as long as the given model can handle backpropagation (no further requirements for layers), and can run much faster than methods with input perturbation. The only overhead of this method is the assumed availability of a latent model over the input. If this latent model is not available, we can learn such a model using generative adversarial methods or variational auto encoders. Learning this latent space needs to be done only a single time and is independent of the learned classifier to be explained.

## 2    RELATED WORK

There are different ways to categorize interpretability methods such as the one discussed by Shrikumar et al. (2017). Here we categorize the existing approaches into three overlapping categories.

### 2.1    NETWORK VISUALIZERS

The first group of methods try to understand units of the network (Erhan et al., 2009; Yosinski et al., 2015; Bau et al., 2017). These methods test each individual unit or a set of units in the network to gain insight into what network has learned.

The disadvantage of these methods is that they need to check all the units to see which one (or combination of units) is responsible for a concept. Morcos et al. (2018) showed it is unlikely that only a single unit learns a concept. Rather, a combination of units usually combined to represent a concept. This, in turn, makes these methods inapplicable in practice when the network contains thousands of units. These methods are example-based explanations. That is, they generate an explanation for a single input. By contrast, Fong and Vedaldi (2018) proposed a method to determine whether the network learned a concept based on a set of probe images and pixel-level annotated ground truth which may not be readily available or easy to obtain for many tasks.

### 2.2    INPUT SPACE VISUALIZERS

The second category corresponds to networks that try to explain the network's decision in the space of the input image. Ribeiro et al. (2016) proposed a method to find out which parts of the image have the largest contribution to the decision of the network by making changes to the image and forwarding the new image through network. Zintgraf et al. (2017) proposed a similar approach with a more clever way of sampling the image parts. These methods need to consider changing each dimension of images and get explanation for each change in order that the aggregated results are visually coherent.

Zhou et al. (2016) proposed a method which forwards the image through the network, records the activations in the last pooling or convolution layer, and uses them as an explanation. Selvaraju et al. (2017) proposed a similar method, but which uses the back-propagated signal for a more accurate explanation. There are two potential difficulties with these approaches. First, they assume that the explanation can be summarized in the last pooling or convolution layer which has a broad receptive field. Explanations with such broad receptive field cannot give useful information in explaining fine-grained datasets. Second, these methods are restricted to use in convolutional networks.

Simonyan et al. (2013) used the gradient of the output with respect to the pixels of the input image to generate a heat map. They showed that their proposed method is closely related to DeconvNets (Zeiler and Fergus, 2014). The difference is in the handling of backpropagation of ReLU units. The weakness of these methods is that the generated backpropagated signal in image space is not visually specific. They need to use heuristics in backpropagation to make the results more specific and useful to humans, such as changing the backpropagation rules for ReLU unit (Springenberg et al., 2015). Kindermans et al. (2018) showed the unreliability of these methods to shifts in input. Some of these methods needs a reference input image (Shrikumar et al., 2017) whose choice can greatly change the generated explanation (Kindermans et al., 2018).

While preparing this manuscript, we discovered xGEMs (Joshi et al., 2018), a preprint available on arXiv. It is similar to our work in that it uses a GAN to regularize the explanation and also seeks to find a "why not" explanation. Their work is different in that it does not formulate a constrained optimization problem as we do (and our results show the importance of our constraints) and they focus on producing a "morph" from one class to another, rather than highlighting the differences directly in the input image (as we do).

### 2.3    JUSTIFICATION EXPLANATIONS

Finally, there are methods that learn to justify the classification of a network by producing textual or visual justifications (Hendricks et al., 2016; Park et al., 2018). Although related to image descriptions or definitions, they differ in that their goal is to explain *why*. However, the justification network is

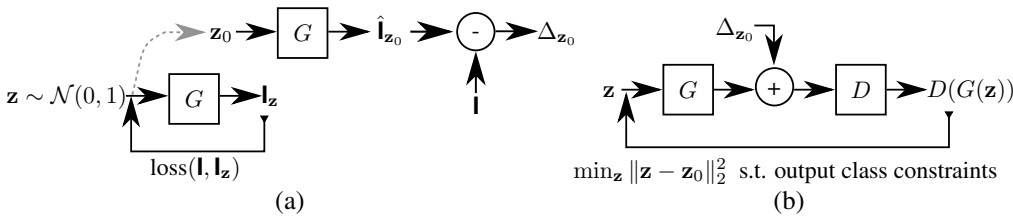

Figure 1: The schematics of the proposed approach. (a) First, find $\mathbf{z}_0$ which gives $G\left(\mathbf{z}_0\right) \approx \mathbf{I}$. Then, let $\Delta_{\mathbf{z}_0}$ to be the difference between the input image $\mathbf{I}$ and reconstructed one, $G\left(\mathbf{z}_0\right)$. (b) Last, generate the final explanation by optimization over $\mathbf{z}$. $\mathbf{z}_0$ and $\Delta_{\mathbf{z}_0}$ are fixed.

trained to match human explanations. Therefore, they are not direct explanations of how the learned classification network makes its decisions, but rather what humans would like as the explanation.

### 2.4 SUMMARY

In summary, the existing approaches in literature have at least one of the following downsides.

I By learning an explanation network, they use another black box to explain a current black box (Park et al., 2018).

II They are not always applicable since they need specific layers or architecture (Zhou et al., 2016; Selvaraju et al., 2017).

III They use heuristics during backpropagation to generate explanations (Springenberg et al., 2015; Zeiler and Fergus, 2014).

IV They need of a set of probe images or concepts which may not be available or cannot be easily obtained (Shrikumar et al., 2017).

V They need network alteration to record the activations (Zeiler and Fergus, 2014; Bau et al., 2017; Shrikumar et al., 2017).

VI They need of considerable amount of computational time to provide an explanation (Zintgraf et al., 2017; Ribeiro et al., 2016).

Our model does not suffer from the aforementioned shortcomings, except that we require a generative model of the input space. Note that this input-space model does not force the explanation. Its purpose is similar to that of reference input images, except (as required by other methods) it does not need any domain knowledge. The input-space model can be estimated entirely independently from the network to be explained.

## 3 PROPOSED METHOD

First, we introduce the notation used in this work in Section 3.1. Then, in Section 3.2, we describe how to learn a latent space capable of generating natural looking images similar to the input space. Last, in Section 3.3, we describe our method on how to generate explanations from the latent representation of input space. The overall framework is summarized in Procedure 1 and Figure 1.

### 3.1 TERMINOLOGY

$D : \mathbb{R}^n \to \mathbb{R}^c$ is the given discriminator network, for which we want to generate explanations. $G : \mathbb{R}^k \to \mathbb{R}^n$ generates natural looking images similar to the ones in the dataset. We choose $G$ to be an adversarial network trained on the input dataset used for $D$. $\mathbf{I} \in \mathbb{R}^n$ is the input image; $\mathbf{z} \in \mathbb{R}^k$ is a latent variable; and $\mathbf{I}_z$ is the output of $G$, i.e. $\mathbf{I}_z = G\left(\mathbf{z}\right)$. $y_{\text{true}}$ is the correct label for image $\mathbf{I}$, and $y_{\text{probe}}$ is the class label for the class of interest.

Thus, the question we would like to answer is "Why did $D$ produce label $y_{\text{true}}$ and not label $y_{\text{probe}}$ for input $\mathbf{I}$?"

---

**Procedure 1** Generating the explanation on given $D$ and $\mathbf{I}$

---

1: Learn a function $G : \mathbb{R}^k \to \mathbb{R}^n$                                                        ▷ See 3.2
2: Find a representation for input $\mathbf{I}$ using procedure 2
3: Find $\mathbf{z}_e$ from Equation 2
4: Return the explanation $G(\mathbf{z}_0) - G(\mathbf{z}_e)$

---

**Procedure 2** Getting latent representation on the input $\mathbf{I}$

---

1: **procedure** LEARN $\mathbf{z}_0$ $(G, \mathbf{I}, \lambda, \text{loss}(.))$
2:      $\mathbf{z}_0 \sim \mathcal{N}(0, 1)$
3:      **while** $G(\mathbf{z}_0) \not\approx \mathbf{I}$ **do**
4:          $\mathbf{z}_0 \leftarrow \mathbf{z}_0 - \lambda \nabla_{\mathbf{z}} \text{loss}(\mathbf{I}, G(\mathbf{z}))$
5:      $\Delta_{\mathbf{z}_0} = G(\mathbf{z}_0) - \mathbf{I}$
6:      **return** $\mathbf{z}_0, \Delta_{\mathbf{z}_0}$

---

## 3.2   LEARNING INPUT DISTRIBUTION

The question "why not class $y_{\text{probe}}$?" implies a query image about which the question is being asked. We need to capture the manifold of natural looking images similar to this input to be able to answer this question in a meaningful way for a human. Learning a compact manifold enables us to move along this manifold instead of directly optimizing on the input space, i.e., each pixel.

There are different ways to find this mapping including variational auto encoders (Kingma and Welling, 2013) and generative adversarial networks (GANs) (Goodfellow et al., 2014). In this work, we used GANs to map latent space into input space. We used the method proposed by Arjovsky et al. (2017) to train the GAN. The structure of the networks is similar to that proposed by Radford et al. (2015).

## 3.3   GENERATING EXPLANATIONS

First, we need to find an initial point in latent space which represents the input image, i.e., $G(\mathbf{z}_0) \approx \mathbf{I}$. We find this initial point by solving $\mathbf{z}_0$ as

$$\mathbf{z}_0 = \arg \min_z \ \text{loss}(G(z), I) \tag{1}$$

in which $\text{loss}(.)$ is a suitable loss function, e.g. $||.||_2$ distance for images. Since the generated image may classified into different class by the discriminator, we add a misclassification cost to our loss function. As the final fit will not be exact, we define $\Delta_{\mathbf{z}_0}$ to be the residual error: $\Delta_{\mathbf{z}_0} = G(\mathbf{z}_0) - \mathbf{I}$. See Procedure 2 for more details.

Next, we find a change in latent space for which the change in the input space explains why, for this particular image, class $y_{\text{probe}}$ is not the answer. In particular, we seek the closest point (in the latent space of $\mathbf{z}$) for which $y_{\text{probe}}$ would be equally likely as $y_{\text{true}}$, and all other classes are less probable. This is a point which is most like the input, yet would be class $y_{\text{probe}}$. Thus, the answer to the question is, "It is *not* like this."

To do so, we solve a constrained optimization problem:

$$
\begin{aligned}
\mathbf{z}_e = \arg \min_{\mathbf{z}} \quad & ||\mathbf{z} - \mathbf{z}_0||_2^2 \\
\text{subject to} \quad & \text{llh}(D(G(\mathbf{z})) + \Delta_{\mathbf{z}_0}, y_{\text{true}}) - \text{llh}(D(G(\mathbf{z})) + \Delta_{\mathbf{z}_0}, y_{\text{probe}}) = 0 \\
& \text{llh}(D(G(\mathbf{z})) + \Delta_{\mathbf{z}_0}, y_{\text{true}}) - \text{llh}(D(G(\mathbf{z})) + \Delta_{\mathbf{z}_0}, y_i) \leq \epsilon, \quad \forall i \neq y_{\text{true}}, i \neq y_{\text{probe}} \\
& \text{llh}(D(G(\mathbf{z})) + \Delta_{\mathbf{z}_0}, y_{\text{probe}}) - \text{llh}(D(G(\mathbf{z})) + \Delta_{\mathbf{z}_0}, y_i) \leq \epsilon, \quad \forall i \neq y_{\text{true}}, i \neq y_{\text{probe}}
\end{aligned}
\tag{2}
$$

in which $\text{llh}(f, y)$ is *log likelihood* of class $y$ for classifier output $f$.

The first constraint forces the explanation to be on the boundary of the two classes. However, because $D$ is complex, this can lead to solutions in which the likelihood of $y_{\text{probe}}$ and $y_{\text{true}}$ are equal, but

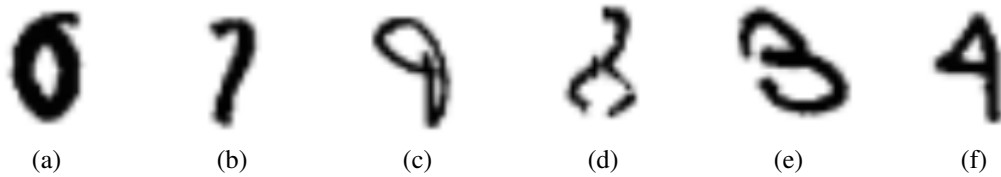

Figure 2: (a) True label is 0 and not 6. (b) True label is 7 and not 1. (c) True label is 9 and not 8. (d) True label is 8 and not 6. (e) True label is 3 and not 9 or 8. (f) True label is 4 and not 9.

another class has an even higher likelihood. To overcome this, we impose the last two constraints which enforce that the true and probe classes remain the most likely. We further illustrate the necessity of these constraints in Section 4.

## 4 EXPERIMENTAL RESULTS

In this section we compare our method, contrastive deep explanation (CDeepEx), with those of Ribeiro et al. (2016); Selvaraju et al. (2017); Zintgraf et al. (2017); Samek et al. (2017). We have tested these methods on two datasets: MNIST (LeCun et al., 1998) and fashion-MNIST (Xiao et al., 2017). Although MNIST seems to be a very simple dataset, it has its own challenges when it comes to contrasting two outputs. This dataset has subtleties that are hard even for humans to grasp, such as the ones depicted in Figure 2.

### 4.1 EXPERIMENTS ON MNIST

In this section, we find explanations for contrasting categories using our method (CDeepEx), Lime (Ribeiro et al., 2016), GradCam (Selvaraju et al., 2017), PDA (Zintgraf et al., 2017), LRP Samek et al. (2017) and xGEMs (Joshi et al., 2018). Our network architecture is similar to that used by the original MNIST papers, consisting of two sets of Conv/MaxPool/ReLU layers following by two fully connected layers. The input is resized to 64x64. The kernel size for each convolution layer is 5x5 and the first fully connected layer is 3380x50.

GradCam and Lime were not designed to answer this type of question directly. Therefore, we generate the explanation by first extracting the explanations for the true and probe classes, and then subtracting a weighted explanation of the probe class from the true class. If imposing this explanation on the input image decreases the network probability for the true class and increases it for the probe class, we keep this explanation. The weights for the probe explanation we used are from 0.005 to 2 with increments of 0.005. The final explanation is the average of all the maps that satisfies the mentioned condition. We tried multiple other methods to contrast the explanations of the true and probe classes and this method produced the best and most reliable results, although clearly GradCam and Lime were not designed for this scenario.

We performed two sets of experiments. In the first set, we trained the discriminator on the unmodified MNIST dataset with a resulting success rate of 99% in testing phase. For testing examples, we asked why the output was not the second-highest likelihood class. (That is the probe class is the second-most likely class, according to the classifier.)

Figure 3 shows explanations generated using CDeepEx, PDA, GradCam, Lime and LRP. The first example shows why it is a 4 and not a 9. Our method shows that this is because the gap on the top of the digit is not closed. The second row is the explanation for why a 0 and not a 6. The only meaningful explanation is generated from CDeepEx: because the circle is thicker and the top extending line is not more pronounced. The third row is for why a 5 and not a 6. Although Lime shows that the opening gap would have to be closed, it produces many other changes that detract from the central reason. For the last row, the only correct explanation for why a 3 and not an 8 comes from our method. Note that GradCam and Lime were not designed for this type of explanation, but we tried hard to find the best way to make contrasting explanations from their single-class explanations (see above).

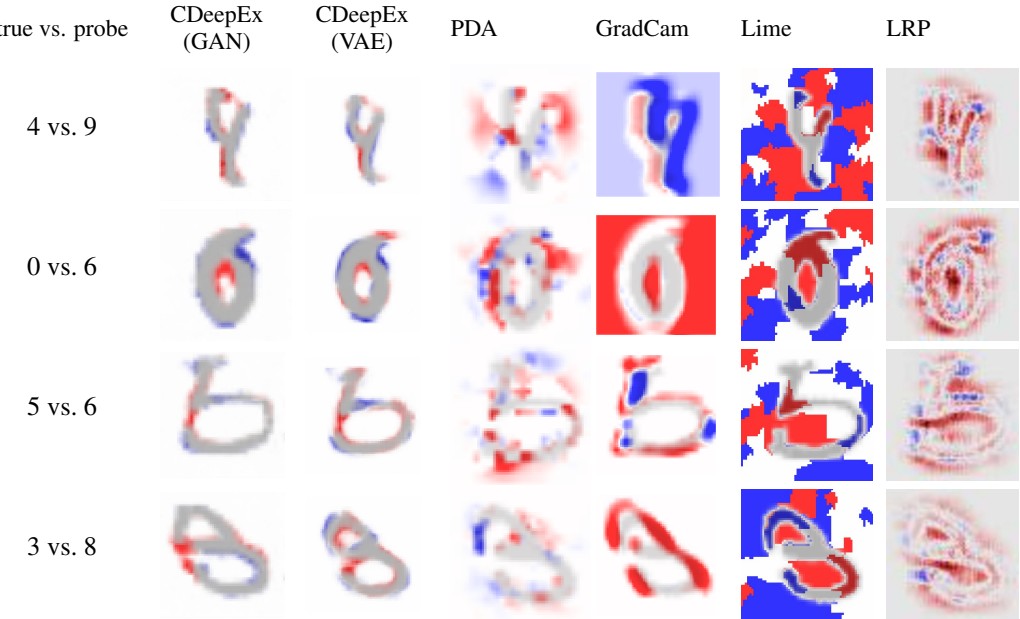

Figure 3: Generated explanations for MNIST on the same trained network. The input image is in gray. Red indicates regions that should be added and blue regions that should be removed (to move from the true/predicted label to the probe label). Thus, the absence of the red regions and the presences of the blue regions explain why the input is the true label and not the predicted label.

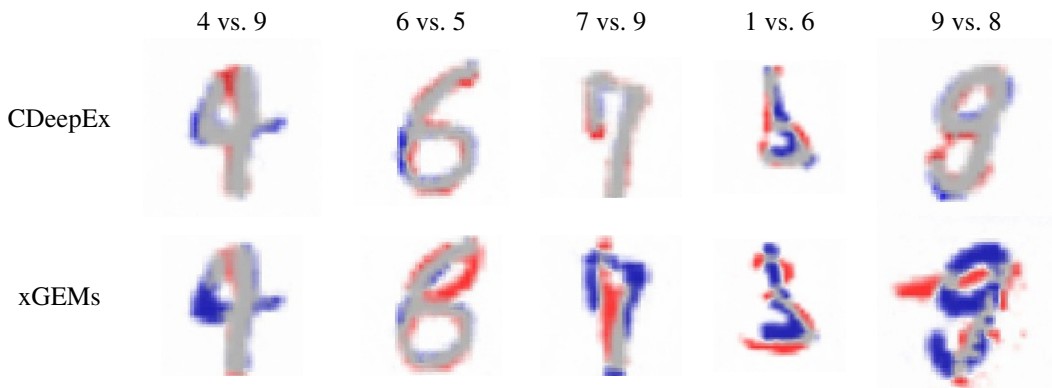

Figure 4: Comparison of CDeepEx to xGEMs. Colors are as in Figure 3.

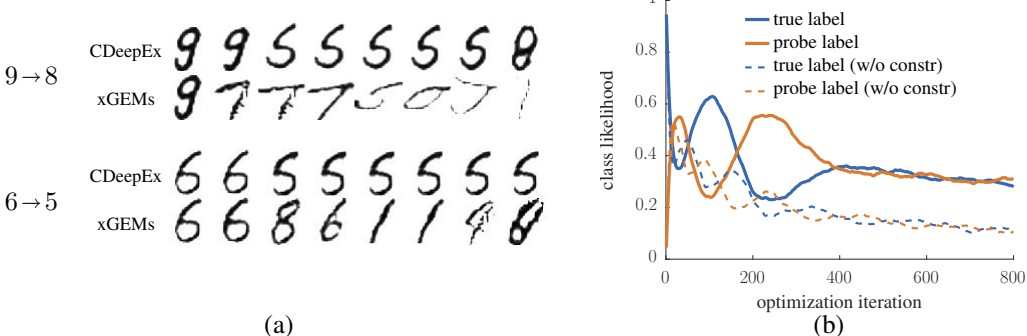

(a) (b)

Figure 5: Examples of the optimization path for Equation 2. CDeepEx is our method. xGEMs is very similar, but without the last pair of constraints. The goal is to find a $\mathbf{z}$ such that $G(\mathbf{z})$ is maximally confused between the two classes. (a) Examples from $G(\mathbf{z})$ along the optimization path of $\mathbf{z}$. (b) Average class likelihood from $D(G(\mathbf{z}))$ for all examples (probe is second-most-likely class). Both demonstrate that without the constraints (xGEMs), the optimization finds an example for which the true and probe likelihoods are equal, but another class has an even higher likelihood. Our method (CDeepEx) with the constraints keeps the explanation targeted to the true and probe classes.

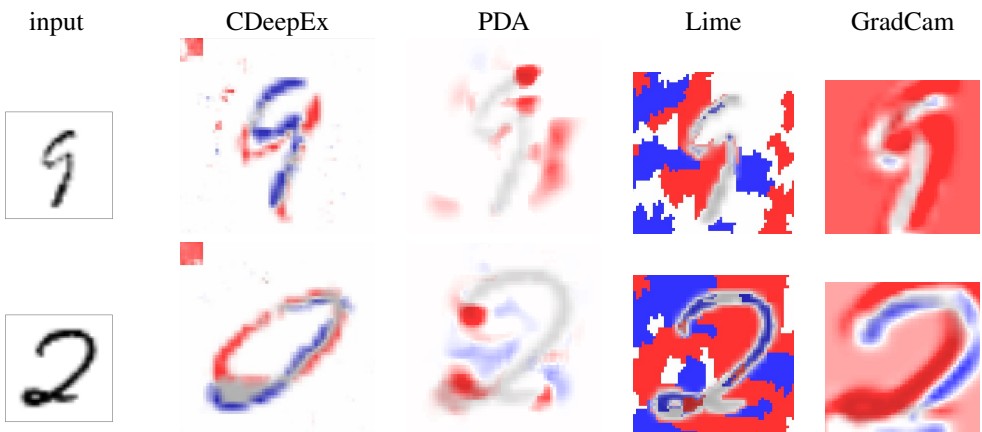

Figure 6: Explanations for "why not 8?" for a network trained on data in which every 8 has a small square in the upper-left corner.

In Figure 4, we compare our results with our implementation of xGEMs. The primary difference between the methods is the last two sets of constraints in Equation 2. This figure shows that without the constraints the explanation are not correct or are of worse quality, compared to explanations with constraints.

To illustrate why, in Figure 5, we show the optimization path to find the explanation, with (CDeepEx) and without (xGEMs) constraints. The program has a non-linear objective with non-linear constraints, so the path is particularly important, as we will not (generally) find the global optimum. We use an augmented Lagrangian method for the optimization (Bertsekas, 1999, Section 4.2). Without the constraints, the found $\mathbf{z}$ results in equal likelihood for $y_{\text{true}}$ and $y_{\text{probe}}$, but a third class consistently has even higher likelihood, thus generating an explanation more suitable to this third class than the requested probe class.

For the second experiment, we trained a network with the same structure as the previous one on the modified data. In this experiment we added a 6x6 gray square to the top left of all the images for class "8." If tested on the true data (without the modifications), the network only recognizes 5% of the 8s correctly. Also, adding the square to all the testing images, the network recognized 77% of other classes as 8.

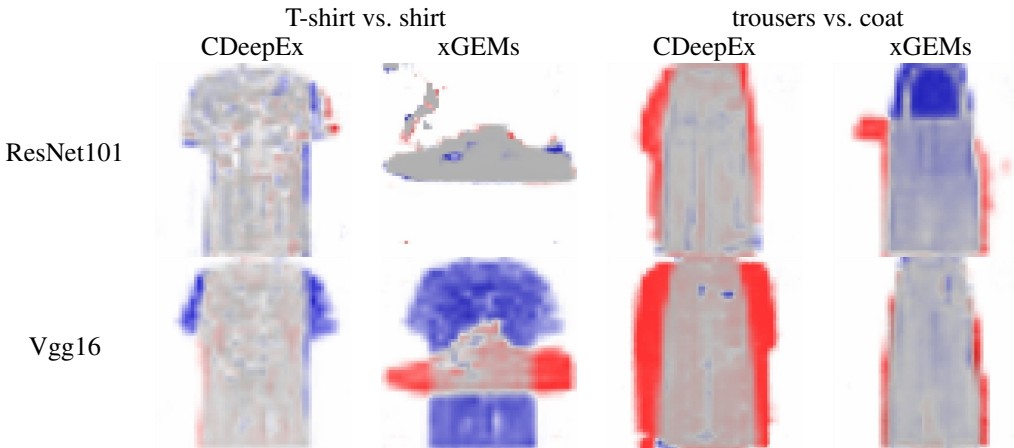

Figure 7: Top and bottom rows are the results for ResNet101 and VGG16 respectively. (a) Change fro T-shirt to a Shirt with constraints. (b) T-shirt to shirt without constraints. (c) Change from trousers to coat with constraints. (d) Change from trousers to coat without constraints.

We then compare our method to others on this same network, trained on biased data. Our results are shown in Figure 6. Our method is clearly able to articulate that adding a square in the upper left makes something an 8, whereas the other methods are not able to find this bias of the network.

## 4.2 FASHION MNIST

We trained two different networks on the Fashion MNIST dataset: Vgg16 (Simonyan and Zisserman, 2014) and ResNet101 (He et al., 2016). The testing accuracy for both networks is 92%. For the generating network, we used the structures and learning method presented by Arjovsky et al. (2017) with latent space of size 200. We then illustrate our method's ability to gain insight into the robustness of the classifiers through constrastive explanations.

We generated explanation with and with out constraints in Figure 7. Comparing CDeepEx with xGEMx, we can see the importance of the constraints. Using CDeepEx, it is clear that Vgg16 learned more general concepts than ResNet101. The first column shows that ResNet101 learned very subtle differences on the surface of the T-shirt to distinguish it from a (long-sleeved) shirt. By contrast, Vgg16 understands removing the short sleeves makes the appearance of a shirt with long sleeves. In the "trousers vs. coat" examples, ResNet101 believes that adding a single sleeve will change the trouser into a coat, while Vgg16 requires both sleeves.

## 5 CONCLUSIONS

Our constrastive explanation method (CDeepEx) provides an effective method for querying a learned network to discover its learned representations and biases. We demonstrated the quality of our method, compared to other current methods and illustrated how these contrastive explanations can shed light on the robustness of a learned network.

Asking a network contrastive questions of the form, "Why is this example not of class B?" can yield important information about how the network is making its decisions. Most previous explanation methods do not address this problem directly, and modifying their output to attempt to answer constrastive questions fails.

Our formulation draws on three ideas: The explanation should be in the space of natural inputs, should be an example that is maximally ambiguous between the true and probe classes, and should not be confused with other classes. The method does not require knowledge of or heuristics related to the architecture or modality of the network. The explanations can point to unintended correlations in the input data that are expressed in the resulting learned network.

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

## 6 SUPPLEMENTARY

In this section we show more qualitative results using both GAN and VAE. In Figure 8, we show the robustness of our method regardless of using a GAN or a VAE. We compared our results against xGEMs. Figure 9 shows how the optimization procedure may find another point that has equal likelihood for probe and the true class but another class has a higher likelihood.

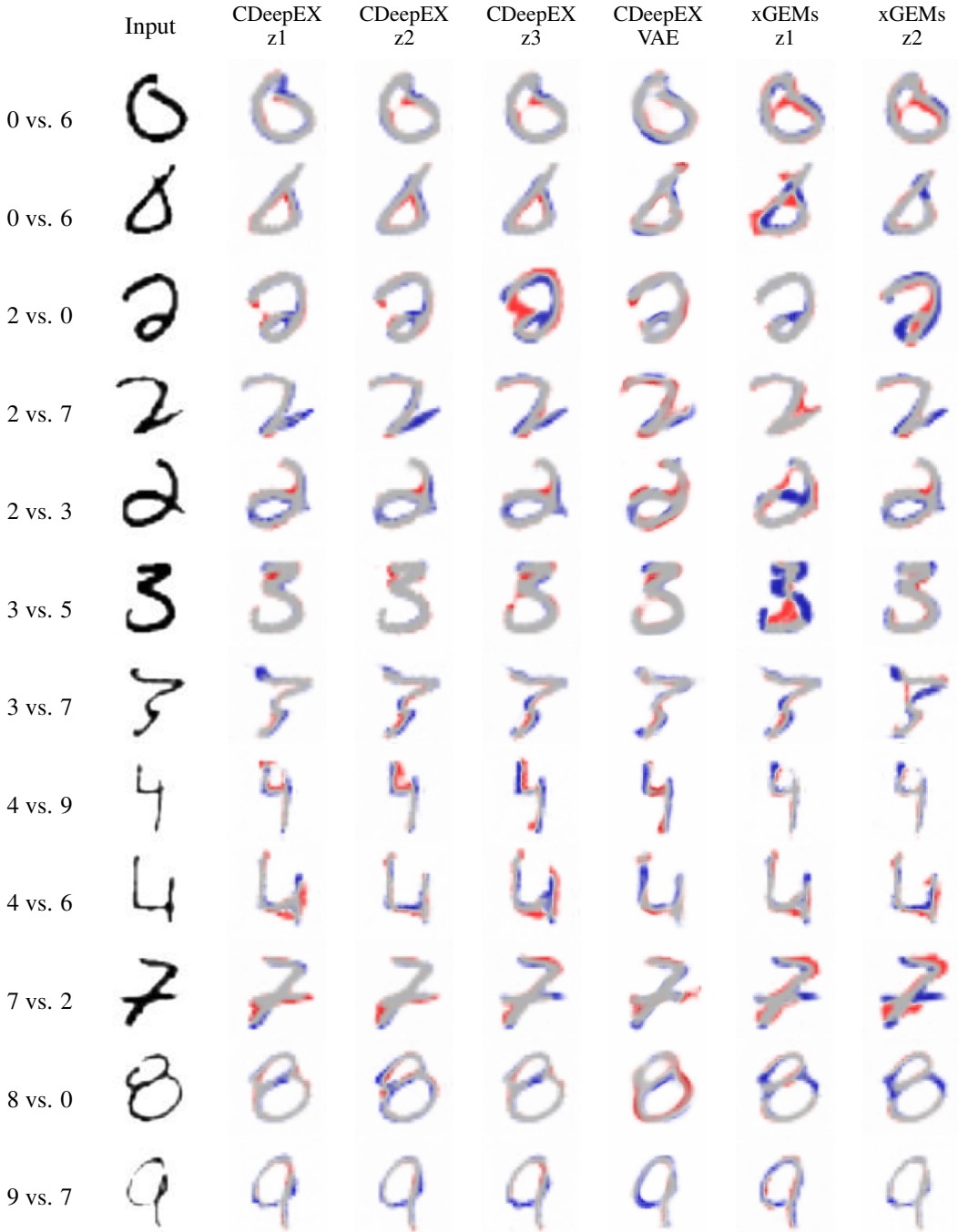

Figure 8: Additional experiments comparing our method using a GAN, using VAE, and xGEMs. The multiple columsn for the GAN methods are for different random starting points for the optimization of z.

Figure 9: Optimization path showing the importance of constraints. In this figure, we used a VAE instead of a GAN. Top row shows CDeepEx optimization path while the bottom row shows xGEMs optimization path.

## 6.1 NETWORKS ARCHITECTURE

The structure of VAE for the MNIST dataset is described below. VAE consists of two parts, a encoder and a decoder. The encoder structure is:
input size: 4096 -> Linear(4096 to 784) -> Linear(784 to 400) -> Linear(400 to 120).
The decoder structure is:
Linear(120 to 400) -> Linear(400 to 784) -> Linear(784 to 4096)

The Discriminator network on MNIST dataset has the following structure:
input size: 4096
Conv2d(1, 10, kernelsize=(5,5), stride=(1,1))
MaxPool2d(kernelsize=(2,2), stride=(2, 2))
ReLU
Conv2d(10, 20, kernelsize=(5, 5), stride=(1, 1))
MaxPool2d(kernelsize=(2, 2), stride=(2, 2))
ReLU
Dropout2d(p=0.5)
Linear(infeatures=3380, outfeatures=50, bias=True)
Linear(infeatures=50, outfeatures=10, bias=True).

The architecture of the generator network used for FMNIST experiments is as follows:
ConvTranspose2d(200, 512, kernel size=(4,4), stride=(1,1), bias=False)
BatchNorm2d(512, eps=1e-05, momentum=0.1, affine=True)
ReLU(inplace)
ConvTranspose2d(512, 256, kernel size=(4,4), stride=(2,2), padding=(1,1), bias=False)
BatchNorm2d(256, eps=1e-05, momentum=0.1, affine=True)
ReLU(inplace)
ConvTranspose2d(256, 128, kernel size=(4,4), stride=(2,2), padding=(1,1), bias=False)
BatchNorm2d(128, eps=1e-05, momentum=0.1, affine=True)
ReLU(inplace)
ConvTranspose2d(128, 64, kernel size=(4,4), stride=(2,2), padding=(1,1), bias=False)
BatchNorm2d(64, eps=1e-05, momentum=0.1, affine=True)
ReLU(inplace)
ConvTranspose2d(64, 1, kernel size=(4,4), stride=(2,2), padding=(1,1), bias=False)
Tanh()

## 6.2 DISCRIMINATOR ACCURACY FOR GENERATING EXPLANATION

We desinged another experiment, showing the explanation results are coming from the discriminator network and not the generative model. We trained a network in the following fashion on MNIST dataset. First, we train the network, without showing it any images from class 8. It drops the testing accuracy on that class to 0. Then, we show some samples of the class 8 to the network, increasing its accuracy over class 8 to 0.3. We repeat this procedure by showing more and more samples to the network, saving its parameters at 0.6 and 0.99 accuracy. Then, we run an experiment asking network to change an input image to class 8. The results are shown at Fig 10.

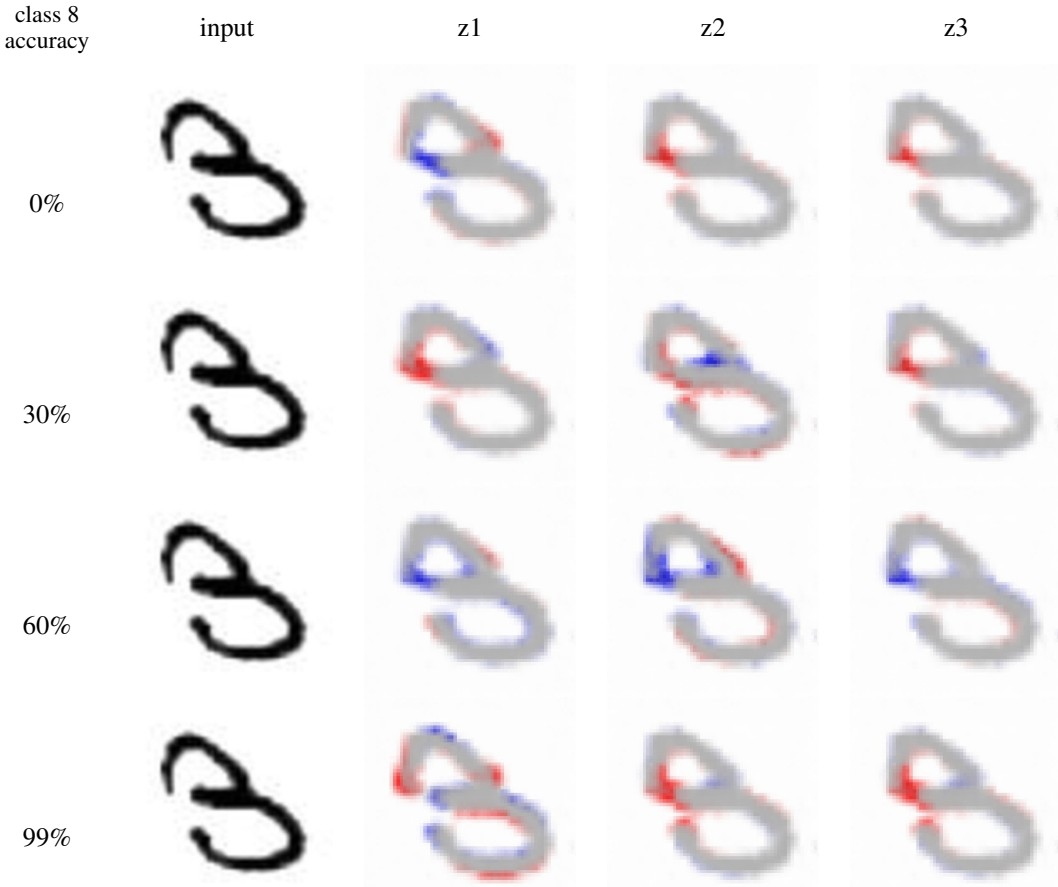

| class 8 accuracy | input | z1 | z2 | z3 |
|---|---|---|---|---|

Figure 10: In this figure we show that the explanation comes from the network we want to speculate, not the generative model. The left column shows the accuracy of the networks over classifying class 8The given discriminator networks have different accuracy for classifying class 8. The second column is the input image. The last three columns are explanations generated using a GAN as generative model, using three different latent codes. The first row shows without the discriminator knowing what is a class 8 instance should look like, the generated explanations are strange and inconsistent. With the first latent code, the network decides to remove the gap from the top curve, while with the other two latent codes, it decides to close the gap. In none of explanations network generates a explanation which wants to close the lower curve. As the accuracy of the network for class 8 increases, the generated explanations are getting better and consistent. Note that increasing the network accuracy does not necessarily adds up linearly to the quality of the generated images (second and third row).

