# OpenReview forum: "CDeepEx: Contrastive Deep Explanations"
_ICLR.cc/2019/Conference_

### Official Review · AnonReviewer2 · 2018-10-29

**Rating:** 5
**Confidence:** 4

**Review:**

The idea proposed in this paper is to aid in understanding networks by showing why a network chose class A over class B.  To do so, the goal is to find an example that is close to the original sample, but belongs to the other class. As is mentioned in the paper, it is crucial to stay on the data manifold for this to be meaningful. In the paper, an approach using a GAN to traverse the manifold is proposed and the experimental evaluation is done on MNIST.

If my understanding is correct the proposed approach requires:
Finding a noise code z_0 such that the GAN generates an image G(z_0) close to the original input x. As a metric L2 distance is proposed.
Find a point close to z_b that is close z_0  s.t. Class B is the most likely class and class A is the second most likely prediction. Specifically it is required that
The log likelihood of but classified as class B with the same log likelihood of class B for G(z_b) is the same as the log likelihood of class A for the input x.
Such that all other classes have a log likelihood that is at least epsilon lower than both the one of class A and class B.

The proposed approach is compared to a set of other interpretability methods, which were
Grad-Cam, lime, PDA, xGEM on MNIST AND Fashion MNIST data. The proposed evaluation is all qualitative, i.e. subjective. It must also be noted that in the methods used for comparison are not used as originally intended.


Currently, I do not recommend this paper to be accepted for the following reasons.
The idea of using a GAN is to generate images in input space is not novel by itself. Although the application for interpretability by counterfactuals is. It is unclear to me how much of the appealing results come from the GAN model and how much come from truly interpreting the network. I have detailed this below by proposing a very simplistic baseline which could get similar results.
The experimental approach is subjective and I am not convinced by the experimental setup.
On the other hand, I do really appreciate the ideas of traversing the manifold.

Remarks
Related work and limitations of existing interpretability methods are discussed properly. Of course, the list of discussed methods is not exhaustive. The work on the PPGAN and the “Synthesizing the preferred inputs for neurons in neural networks via deep generator networks” is not mentioned although it seems very related to the proposed approach to traverse the manifold. What that work sets apart from the proposed approach is that is could be applied to imaganet and not just MNIST.

Traversing the manifold to generate explanations is certainly a good idea and one that I completely support. One limitation of the proposed approach is that it is unclear to me whether a point on the decision boundary is desirable or that a point that is equally likely is desirable. My reasoning is that the point on the decision boundary is the minimal change and therefore the best explanation. In such a setup, the GAN is still crucial to make sure the sample remains on the data manifold and is not caused by adverarial effects.

The exact GAN structure and training approach should be detailed in this paper. Now only a reference is provided.

Can you clarify how the constraints are encoded in the optimization problem?

The grad cam reference has the wrong citation

I do not understand the second paragraph of section 4.1. As mentioned in the paper, these other methods were not designed to generate this type of application. Therefore the comparison could be considered unfair.

I would propose the following baseline. For image x from class A, find image y from class B such that x-y has minimal L2 norm and is correctly classified. Use y instead of the GAN generated image. Is the result much less compelling? Is it actually less efficient that the entire GAN optimization procedure on these relatively small datasets?


I do have to say that I like the experiment with the square in the upper corner. It does show that the procedure does not necesarrily exploits adversarial effects. However, the baseline proposed above would also highlight that specific square?


Figure 5 shows that multiple descision boundaries are crossed. Is this behaviour desired? It seems very likely to me that it should be possible to move from 9 to 8 while staying on the manifold without passing through 5? Since the method takes a detour through 5’s is this common behaviour?


FINAL UPDATE
--------------------
Unfortunately, I am not entirely convinced by the additional experiments that we are truly looking into the classifier instead of analyzing the generative model.
I believe this to be currently the key issue that, even after the revision, needs to be addressed more thoroughly before it can be accepted for publication.

---

> ### Author Response · Authors · 2018-11-22
> **Answers part 1**
>
>
> Comment: The idea of using a GAN is to generate images in input space
> is not novel by itself.  Although the application for interpretability
> by counterfactuals is.  It is unclear to me how much of the appealing
> results come from the GAN model and how much come from truly interpreting
> the network.
>
> Response: We do not believe we claimed generating images in input space
> as a novel contribution of the work.  We have added results using a VAE
> (instead of a GAN) to demonstrate that the GAN is not providing the
> explanation power; it is merely serving to keep the changes in the space
> of natural images.
>
>
> Comment: One limitation of the proposed approach is that it is unclear
> to me whether a point on the decision boundary is desirable or that a
> point that is equally likely is desirable.  My reasoning is that the
> point on the decision boundary is the minimal change and therefore the
> best explanation. In such a setup, the GAN is still crucial to make sure
> the sample remains on the data manifold and is not caused by adverarial
> effects.
>
> Response: Perhaps we have misunderstood the comment.  We are selecting
> a point on the boundary between two classes (probe and true).  Among all
> of the points on the boundary between the two classes, we select the one
> of minimal change to the input (image).  Is the suggestion for something
> different?
>
>
> Comment: The exact GAN structure and training approach should be detailed
> in this paper. Now only a reference is provided.
>
> Response: We have added these details to the supplementary materials. Thank
> you for your suggestion.
>
>
> Comment: Can you clarify how the constraints are encoded in the
> optimization problem?
>
> Response: We encoded them mathematically as in the paper.  We solved them
> as detailed in Bertsekas, D. P. (1999). Nonlinear Programming. Athena
> Scientific, second edition, section 4.2.  It is an alternating optimization
> problem.  One part has a closed form.  The second part can be performed
> by using backpropagation through the network.
>
> Comment: The grad cam reference has the wrong citation.
>
> Response:  Thank you.  We have fixed this.
>
>
> Comment: I do not understand the second paragraph of section 4.1. As
> mentioned in the paper, these other methods were not designed to generate
> this type of application. Therefore the comparison could be considered
> unfair.
>
> Response:  The only method we found before submitting the paper which was
> able to answer the contrastive explanation was xGems.  However, other
> methods could be shoe-horned into trying to answer the question of "why A
> and not B?" and so we figured we should demonstrate that they were not
> sufficient and that a new method (like ours) was necessary.

---

> > ### Comment · AnonReviewer2 · 2018-11-24
> > **Thanks**
> >
> > Thanks you for your clarifications.
> >
> > You argue, gradient descent on the input image is not providing a satisfactory explanation because of adversarial effects, and I agree.
> > However even with a VAE or GAN, it is unclear how much of the explanation comes from what the network thinks vs what the generative model thinks.
> >
> > I would highly recommend not abusing other methods in a comparison. It is fine to discuss their limitations and argue for the need for a new approach. However, it would be more useful to use the space now devoted to this skewed comparison to highlight the effectiveness of the proposed approach.

---

> > > ### Author Response · Authors · 2018-11-26
> > > **Explanations do not come from Generative  models**
> > >
> > >
> > > comment: You argue, gradient descent on the input image is not providing a satisfactory explanation because of adversarial effects, and I agree.
> > > However even with a VAE or GAN, it is unclear how much of the explanation comes from what the network thinks vs what the generative model thinks.
> > >
> > > Response: We added another section to our supplementary materials, showing the effect of generative models on the explanations.

---

> ### Author Response · Authors · 2018-11-22
> **Answers part 2**
>
>
>
> Comment: I would propose the following baseline. For image x from class
> A, find image y from class B such that x-y has minimal L2 norm and is
> correctly classified. Use y instead of the GAN generated image. Is the
> result much less compelling? Is it actually less efficient that the entire
> GAN optimization procedure on these relatively small datasets?
>
> Response: This is an interesting baseline but not related to the task.
> This baseline asks for, ”what is a good change in input image based
> on the training samples which are correctly classified?” This
> is fundamentally different from asking ”what does the *network*
> think is a good change to input image?” We are not looking for the
> "correct" explanation.  We are looking for what the network thinks
> that is a correct explanation, which may be right or wrong.  Consider a
> network trained on MNIST with 99% accuracy.  This says that the network
> correctly classifies almost all of the images from the probe class.  Then,
> the baseline reduces to which image from probe class has the minimum L2
> distance to the input image.  In this procedure, the trained network has
> almost no role, and therefore it is not an explanation of the network,
> just an explanation of the data.
>
>
> Comment: Figure 5 shows that multiple descision boundaries are crossed. Is
> this behaviour desired? It seems very likely to me that it should be
> possible to move from 9 to 8 while staying on the manifold without
> passing through 5? Since the method takes a detour through 5s is this
> common behaviour?
>
> Response: It is not about desirability. This figure shows that without
> adding constraints, the method may go through other parts of manifold,
> resulting in a wrong speculation image.  That is, xGEMs will not find a
> point on the boundary between the two desired classes, but somewhere else
> (where the two classes have equal probability, but lower than some third
> class).  The classification space is complicated enough that "staying near"
> the input is not sufficient.
>
> Figure 5a, top shows that the network knows about the importance of the
> curves, if we impose our constraints.  If we remove the constraints (like
> in xGEMs), we lose this explanation and revert to something meaningless,
> as the optimization path explores other classes.
>
> Note that these are just paths of the optimizer (only the point at the end
> of the optimization path is the "answer").  However, they demonstrate the
> difficulty with optimization in this complex decision space.  Figure 5b
> shows the problem, where xGEMs (with the constraints) fails to keep the
> answer in the area of high likelihood for either of the classes.

---

> > ### Comment · AnonReviewer2 · 2018-11-24
> > **Clarification**
> >
> > Thank you for your response, I should have been more clear in my original review.
> >
> > The reason I suggested the additional baseline is that I expect it to provide results of similar (visual) quality to the proposed approach.
> >
> > Therefore, the important issue is:
> > "Does the proposed approach truly visualize what the network thinks?"
> >
> > The approach consists of two parts. The first part is gradient descent through the network, "i.e. what the network thinks", the second part is continuing this all the way through the GAN to generate the right image.
> >
> > If we would not use the GAN, but use standard gradients on the images, we can find an image that has even smaller L2 distance. However, I think that most of us would agree that these adversarial approaches do not visualise what the network thinks.
> >
> > Now the question becomes: why does the GAN visualise what the network thinks, where gradient descent does not?
> > Assume the following generative network is used (fully connected network):
> > - 100 dimensional code
> > - The weights are the top 100 PCA directions.
> > We know that we can implement PCA using an auto-encoder. So this is a neural network generative model.
> >
> > While the auto-encoder is not as good as a GAN as a generative model, it is unclear to me why would this approach would be worse at visualizing what the network thinks.  It is crucial that this aspect can be experimentally validated.
> >
> > (There is a similar problem with deep-dream style approaches: https://distill.pub/2017/feature-visualization/ the latest iterations are generating amazing pictures, but it is unclear how much of this comes from the network and how much comes from the other tricks).
> >
> > Finally, if the L2 distance is small this does not mean that we are close by, especially as we increase the number of dimensions. Consider https://openreview.net/pdf?id=S1xoy3CcYX figure 5. They have visualised two images with similar L2 distance, however "small" L2 does not mean are similar. Therefore I believe it is crucial how you traverse space and that this path is short (for an appropriate measure of length) and does not traverse multiple decision boundaries.

---

> > > ### Author Response · Authors · 2018-11-26
> > > **Other generative models**
> > >
> > >
> > > comment: While the auto-encoder is not as good as a GAN as a generative model, it is unclear to me
> > > why would this approach would be worse at visualizing what the network thinks.  It is crucial that
> > > this aspect can be experimentally validated.
> > >
> > > Response: We have added experiments using VAE, showing the results do not change noticeably. This suggests that the
> > > explanations generated by our method come from the discriminator
> > > and not the generative model itself. Also, we added another section to our supplementary materials,
> > > showing the effect of generative models on the explanations.
> > >
> > > comment: Finally, if the L2 distance is small this does not mean that we are close by, especially as we
> > > increase the number of dimensions. Consider https://openreview.net/pdf?id=S1xoy3CcYX figure 5.
> > > They have visualised two images with similar L2 distance, however "small" L2 does not mean are similar.
> > > Therefore I believe it is crucial how you traverse space and that this path is short (for an appropriate
> > > measure of length) and does not traverse multiple decision boundaries.
> > >
> > >
> > > Response: We agree. Yet, the L2-norm gives us an smooth transition. Traversing multiple decision boundaries is
> > > not necessarily a disadvantage. Also experiments on xGEMs shows that minimizing the length of this optimization
> > > may not lead to something sensible in the image space. One may conclude that the network has not learned
> > > related concepts, while using our method shed more light on what concepts network has learned.

---

### Official Review · AnonReviewer1 · 2018-11-01
**interesting idea with potential**

**Rating:** 6
**Confidence:** 5

**Review:**


The paper addresses the problem of providing saliency-based visual explanations of deep models tasked at image classification. More specifically, instead of generating visualizations directly highlighting the image pixels that support the the decision of an image belonging to class A, it generates "contrastive" visualizations indicating the pixels that should be added or suppressed in order to support the decision of a image belonging to class A and not to class B.

The method formulates the generation of these contrastive explanations through a generative adversarial network (GAN), where the discriminator D is the image classification model to be explained and the generator G is a generative model trained to produce images from the dataset used to train D.

Experiments on the MNIST and fashion-MNIST datasets compares the performance of the proposed method w.r.t. some methods from the literature.


Overall the manuscript is well written and its content is relatively easy to follow. The idea of generating contrastive explanations through a GAN-based formulation is well motivated and seems novel to me.

My main concern with the manuscript are the following:

i) The proposed method seems to be specifically designed for the generation of contrastive explanations, i.e. why the model predicted class A and not class B. While the generation of this type of explanations is somewhat novel, from the text it seems that the proposed method may not be able to indicate what part of the image content drove the model to predict class A. Is this indeed the case?

ii) Although the idea of generating contrastive explanations is quite interesting, it is not that novel. See Kim et al., NIPS'16, Dhurandhar et al., arXiv:1802.07623. Moreover, regarding the presented results on the MNIST dataset (Sec 4.1) where some of the generated explanations highlight gaps to point differences between digit classes. The work from Samek et al., TNNLS'17 and  Oramas et al., arXiv:1712.06302 seem to display similar properties in their explanations without the need of explicit constractive pair-wise training/testing. The manuscript would benefit from positioning the proposed method w.r.t. these works.

iii) Very related to the first point, in the evaluation section (Sec.4.1) the proposed method is compared against other methods in the literature. Three of these methods, i.e. Lime, GradCam, PDA, are not designed for producing contrastive explanations, so I am not sure to what extend this comparison is appropriate.

iv) Finally, the reported results are mostly qualitative. I find the set of provided qualitative examples quite reduced. In this regard, I encourage the authors to update the supplementary material in order to show extended qualitative results of the explanations produced by their method.
In addition, I recommend complementing the presented qualitative comparisons with quantitative evaluations following protocols proposed in existing work, e.g. a) occlusion analysis (Zeiler et al., ECCV 2014, Samek et al.,2017), a pointing experiment (Zhang et al., ECCV 2016), or c) a measurement of explanation accuracy by feature coverage (Oramas et al. arXiv:1712.06302).

---

> ### Author Response · Authors · 2018-11-22
> **Answers**
>
> Comment: The proposed method seems to be specifically designed for the
> generation of contrastive explanations, i.e.  why the model predicted class
> A and not class B. While the generation of this type of explanations is
> somewhat novel, from the text it seems that the proposed method may not
> be able to indicate what part of the image content drove the model to
> predict class A. Is this indeed the case?
>
> Response: The goal of this paper is not to answer "why A?" but rather
> "why A and not B?"  The visual answer to the two questions may be similar,
> but it may not.  We seek to highlight what in the image would need to
> change to make it a B and not an A, as a way of explaining this contrast.
> There are other papers that seek to answer the question of "why A?" but
> that is not our focus.
>
>
> Comment: Although the idea of generating contrastive explanations is
> quite interesting, it is not that novel. See Kim et al., NIPS’16,
> Dhurandhar et al., arXiv:1802.07623.
>
> Response: Dhurandhar et al. does use the term constrastive explanation.
> However, they look at the question of "why A?"  Contrastive in their case
> refers to whether something is or is not present that drives the
> classification of "A."  This is a different constrast than ours that
> contrasts "A" to "B" (rather than "present for A" to "absent for A").
> We think this is also an interesting form of explanation, but a different
> one.
>
> Kim et al. also has a different form of model criticism; they look at the
> dataset as a whole for examples that help explain.  We look at a different
> problem: for a given example (perhaps not even from the training set),
> why is it not class B?
>
>
> Comment: The work from Samek et al., TNNLS’17 and Oramas et al.,
> arXiv:1712.06302 seem to display similar properties in their explanations
> without the need of explicit constractive pair-wise training/testing. The
> manuscript would benefit from positioning the proposed method w.r.t. these
> works.
>
> Response: The work from Samek et al. is similar to PDA in its essence. We
> will add the comparison with this method to our work for sake of
> completeness.  In the experiment section of Oramas et al., they proposed
> a synthetic flowers dataset that can be used for our purpose.  Since it
> is synthetic and fine-grained, we can compare the method qualitatively
> and quantitatively.  We sent a request to the authors for accessing the
> dataset.  If we granted this access we will add quantitative comparisons
> to our paper.
>
>
> Comment: In the evaluation section (Sec.4.1) the proposed method is
> compared against other methods in the literature. Three of these methods,
> i.e. Lime, GradCam, PDA, are not designed for producing contrastive
> explanations, so I am not sure to what extend this comparison is
> appropriate.
>
> Response: The only method we found before submitting the paper which was
> able to answer the contrastive explanation was xGems.  However, other
> methods could be shoe-horned into trying to answer the question of "why A
> and not B?" and so we figured we should demonstrate that they were not
> sufficient and that a new method (like ours) was necessary.
>
>
> Comment: The reported results are mostly qualitative. I find the set of
> provided qualitative examples quite reduced. In this regard, I encourage
> the authors to update the supplementary material in order to show extended
> qualitative results of the explanations produced by their method.
>
>
> Response: We have added a supplementary section, adding more qualitative
> results. Thank you for your suggestion.

---

### Official Review · AnonReviewer3 · 2018-11-02
**Interesting idea but lacks experimental justification**

**Rating:** 5
**Confidence:** 4

**Review:**

The paper proposes an approach to provide contrastive visual explanations for deep neural networks -- why the network assigned more confidence to some class A as opposed to some other class B. As opposed to the applicability of previous approaches to this problem -- the approach is designed to directly answer the contrastive explanations question rather adapting other visual saliency techniques for the same. Overall, while I find the proposed approach simple -- the paper needs to address some issues regarding the claims made and should provide more quantitative experimental results justifying the same.

- Apart from some flaws in the claims made in the paper, the paper is easy to follow and understand.
- Assuming the availability of a latent model over the images of the input distribution, the proposed approach is directly applicable and faster.
- The authors clearly highlight the problems associated with existing explanation modalities and approaches; ranging from ones applicable to only specific deep architectures to ones using backpropagation based heuristics.
- The proposed approach to generate contrastive explanations is simple and is structured along the lines of methods utilizing probe images to explain decisions -- except for the added advantage that the provided explanations are instance-agnostic due to the assumption of a latent model over the input distribution.

Comments:
- One of the problems highlighted in the paper regarding existing explanation modalities is the use of another black-box to explain the decisions of an existing deep network (also somewhat of a black-box) which the authors claim their model does not suffer from. The proposed approach provides explanations by operating in the latent space of a learned generative model of the input distribution. The learned generator in itself is somewhat of a black-box itself -- there has been prior work indicating how much of the input distribution are GANs able to capture. As such, conditioning on a generative model to propose such contrastive explanations is to some extent using another black-box (generator) to explain the decisions of an existing one. Thus, the above claim made in the paper does not seem well-founded. Furthermore, in experiments, the paper does not provide any quantitatively convincing results to suggest the generator in use is a good one.
- While the authors suggest that a latent model over the input distribution needs to be trained only once and is applicable off-the-shelf for any further contrastive explanations regarding any network operating on the same dataset -- learning such a model of the input space is an overhead in itself. In this light, experiments demonstrating comparisons between GANs and VAEs as the reference generative model for explanations would have made the paper stronger (as the proposed approach relies explicitly on how good the generative model is).
- The paper proposes an interesting experiment to show that the proposed approach is somewhat capable of capturing slightly adversarial biases in the input domain (adding square to the top-left of images of class ‘8’). While I like this experiment, I feel this has not been explored to completion in the sense of experimenting with robustness with respect to structured as well as unstructured perturbations.
- Typographical Errors: Section 3.1 repeats the use of D for a discriminator as well as the input distribution. Procedure 1 and Procedure 2 share the same titles -- which is slightly misleading. In addition, Procedure 1 is not referenced in the text which makes is hard to understand the utility of the same. In Section 4.1, the use of Gradcam and Lime to generate counterfactual explanations is not very clear and makes it slightly hard to follow. Citations used for Gradcam are wrong -- Sundarajan et al., 2016 should be changed to Selvaraju et al., 2017.

Experimental Issues:
- Experimental results are provided only on MNIST and Fashion-MNIST. Since the paper focuses explicitly on providing contrastive explanations for choosing a class A over another class B -- experiments on datasets which do not have real-images seem insufficient. Additional experiments on at least ImageNet would have made the paper stronger.
Regarding contrastive explanations, experiments on datasets where distractor classes (y_probe) are present in addition to the class interest (y_true) seem important -- PASCAL VOC, COCO, etc. Specifically, since the explanations provided are visual saliency maps the paper would have been stronger if there were experiments suggesting -- what needs to change in a region of an image classified as a ‘cat’ to be classified as a ‘dog’ while there is an instance of the class - ‘dog’ present in the image itself. Also, section 7 in Gradcam (https://arxiv.org/pdf/1610.02391.pdf) provides a procedure to generate counter-factual explanations using Gradcam. Is there a particular reason the authors did not choose to adopt the above technique as a baseline?
- Experimental results provided in the paper are only qualitative -- as such, I do not find the comparisons (and improvements) over the existing approaches convincing enough. Since, there is no clear metric to evaluate contrastive explanations -- human studies to judge the class-discriminativeness (or trust) of the proposed approach would have made the paper stronger.

The authors adressed the issues raised/comments made in the review. In light of my comments below to the author responses -- I am not inclined towards increasing my rating and will stick to my original rating for the paper.

---

> ### Author Response · Authors · 2018-11-22
> **Answers**
>
> Comment: One of the problems highlighted in the paper regarding existing
> explanation modalities is the use of another black-box to explain the
> decisions of an existing deep network (also somewhat of a black-box) which
> the authors claim their model does not suffer from.
>
> Response: While the GAN is certainly another black box, it is a function
> just of the data (or the data domain), and not a function of the
> discriminator from which we want to extract explanations.  Therefore,
> training different models, switching prediction tasks on the same
> domain, or any other similar changes would not require changing the GAN.
>
> Comment: Learning such a model of the input space is an overhead in itself.
>
> Response: Overhead calculations of some form are almost impossible to
> avoid.  Whether this is an overly large computational burdon will depend
> on the problem, although we believe the GAN or VAE need only be trained once
> per domain.
>
> Comment: The paper does not provide any quantitatively convincing results
> to suggest the generator in use is a good one.
>
> Response: Measuring the reconstruction accuracy (of goodness of the
> generator) is difficult, as each measure has its own flaws.  For instance,
> Norm measures are sensitive to translations in the image.
>
>
> Comment: Experiments demonstrating comparisons between GANs and VAEs as the
> reference generative model for explanations would have made the paper
> stronger (as the proposed approach relies explicitly on how good the
> generative model is)
>
> Response: This is a good suggestion. We have added experiments using
> with variational autoencoders (VAEs) instead of GANs.  We believe these
> address this concern and some of the comments above.  Thank you.
>
>
> Comment: The paper proposes an interesting experiment to show that the
> proposed approach is somewhat capable of capturing slightly adversarial
> biases in the input domain (adding square to the top-left of images of
> class 8). While I like this experiment, I feel this has not been explored
> to completion in the sense of experimenting with robustness with respect
> to structured as well as unstructured perturbations.
>
> Response: While we could certainly perform more experiments in this vein,
> we are uncertain what type of unstructured perturbations would be useful
> (and how then to measure whether the technique captured the correct
> explanation).
>
>
> Comment: typographical errors...
>
> Response:  Thank you.  We have fixed the typos.
>
>
> Comment: Section 7 in Gradcam (https://arxiv.org/pdf/1610.02391.pdf)
> provides a procedure to generate counter-factual explanations using
> Gradcam. Is there a particular reason the authors did not choose to adopt
> the above technique as a baseline?
>
> Response: The the proposed counter-factual experiment for GCAM produces
> *any* counter-factual explanation, not a targeted explanation.  It answers
> "why A?" and not "why A and not B?" as we do in this paper.

---

> > ### Comment · AnonReviewer3 · 2018-11-29
> > **Comments on Author Response**
> >
> > Thanks to the authors for addressing the concerns raised in the reviews, responding appropriately and updating the paper to reflect the same wherever applicable.  In light of the comments below (regarding author responses to review comments and clarifications), I am not inclined towards increasing my rating. I will be sticking to my original rating for the paper.
> >
> > - Response: “While the GAN….changing the GAN”.
> > Comment: I agree with the authors that once the generator has been trained on the data-domain, it need not be changed to generate contrastive explanations from a model in play. My point here was the degree of error to which the generator has been learned is going to be carried over to any form of contrastive explanations extracted from tasks on the same domain. Learning a perfect generator in this regard is a hard task in itself -- and could depend significantly on the dataset being operated upon.
> >
> > - Response: “ Learning such …. Itself”
> > Comment: Again, agree with the fact that the GAN/VAE need not be trained again on the same data-domain. The authors suggest that overhead calculations are almost impossible avoid. In context of generating explanations, I do not agree with this statement -- while not for counterfactual explanations Grad-CAM does avoid any learning component on top of the model (to be explained). And since there are no quantitative experiments justifying improvements over the methods being compared with -- it seems like the claim made here does not have significant backing w.r.t the problem of generating contrastive explanations.
> >
> > - Response: “Measuring ….image.”
> > Comment: I agree the metrics in place for evaluating likelihood-based or point-sample based generative models may not be truly reflective of what is being evaluated but having a sense of where these metrics lie for the generator being used here would have made the paper stronger.
> >
> > In addition, as mentioned in my review above -- with respect to contrastive explanations, experiments on datasets where distractor classes (y_probe) are present in addition to the class interest (y_true) seem important to me.

---

### Author Response · Authors · 2018-11-22
**General Comment**



We thank the reviewers for their comments and reviews.  We address many
of the comments below, including by adding additional experiments, as
suggested.

We would like to stress again that the purpose of the method is
to determine how the neural network is making decisions, *not* necessarily
to find general distinctions in the data, although the two
are certainly related.

We have submitted a revised version of our paper. This revision contains
the following:

1 - Experiments with VAE instead of GAN, showing the robustness of our approach.
2 - Adding comparisions with Layerwise Relevance Propagation (LRP)
3 - Adding experiments in supplementary section using VAE instead of GAN
using xGem method, showing the importance of the constraints regardless
of the model being used.
4 - adding more qualitative samples in the supplementary section.
5 - The structure of the networks we have used.

---

### Meta-Review · Area_Chair1 · 2018-12-17

**Confidence:** 5
**Recommendation:** Reject

**Metareview:**

Paper studies an important problem -- producing contrastive explanations (why did the network predict class B not A?). Two major concerns raised by reviewers -- the use of one learned "black-box" method to explain another and lack of human-studies to quantify results -- make it very difficult to accept this manuscript in its current state. We encourage the authors to incorporate reviewer feedback to make this manuscript stronger for a future submission; this is an important research topic.